# Reasons for Non-Attendance in the German National Mammography Screening Program: Which Barriers Can Be Overcome Using Telephone Counseling?—A Randomized Controlled Trial

**DOI:** 10.3390/healthcare11010017

**Published:** 2022-12-21

**Authors:** Sebastian Fochler, Kerstin Weitmann, Martin Domin, Wolfgang Hoffmann

**Affiliations:** 1Department of Diagnostic Radiology and Neuroradiology, University Medicine Greifswald, 17475 Greifswald, Germany; 2Institute for Community Medicine, University Medicine Greifswald, 17475 Greifswald, Germany

**Keywords:** mammography, screening, non-attendance, telephone consultation

## Abstract

Introduction: Germany has established a national mammography screening program (MSP). Despite extensive awareness campaigns, the participation rate is only 54%, which is considerably below the European guidelines’ recommendation of at least 70%. Several reasons why women do not participate are already known. Telephone consultations along with invitation letters have improved the participation rate. Here, we analyzed the reasons for non-participation and offered barrier-specific counseling to examine which impediments can be overcome to improve participation. Study Design: In a randomized controlled trial, women who had not attended their proposed screening appointment in the MSP after a written invitation were contacted by telephone and asked why they did not attend. Barrier-specific counseling via telephone was then offered. Participation in the MSP was rechecked 3 months after counseling. Setting: 1772 women, aged 50–69 years, who had not scheduled a mammography screening after a written invitation were contacted by telephone and asked for their reasons for non-participation. Intervention: The reasons were recorded by the calling consultant and categorized either during the call or later based on their recorded statements. Afterward, the women received counseling specific to their statements and were given general information about the MSP. Main outcome measures: We categorized the reasons given, calculated their frequency, and analyzed the probabilities to which they could be successfully addressed in individual counseling. Participation rates were determined post-consultation according to the reason(s) indicated. Results: The data were analyzed in 2022. After exclusions, 1494 records were analyzed. Allowing for multiple reasons to be stated by every individual 3280 reasons for not attending were abstracted. The most frequent reason was participation in “gray screening” (51.5%), which included various breast cancer prevention measures outside the national MSP. Time problems (26.6%) and health reasons (17.3%) were also important. Counseling was most effective when women had not participated for scheduling reasons. Conclusion: Several reasons prevented women from participating in the MSP. Some reasons, such as time-related issues, could be overcome by telephone counseling, but others, like barriers resulting from fear of the examination procedure or its result, could not.

## 1. Introduction

Mammography screenings can effectively reduce breast cancer mortality [1,2,3,4,5,6]. In Germany, a national mammography screening program (MSP) was gradually introduced between 2005 and 2008 and is now implemented in all German states. All women aged 50–69 years are invited in biennial intervals to receive a mammogram. Personal data on the invitees are derived from local registries. The cooperative association for mammography confirmed the success of the MSP in 2009, demonstrating that a higher rate of in situ carcinomas was detected via the MSP compared with that before 2009 [7]. A higher rate of in situ carcinomas in a screening mammogram is considered a surrogate parameter for breast cancer mortality reduction [7]. The MSP is implemented based on the European guidelines for the implementation of mammography screenings [8]. These guidelines recommend the desired participation rates, technical specifications, and quality assurance measures. Despite the efforts of state organizations and individual mammography units to inform women about the dangers of breast cancer and the benefits of early detection, the participation rate is 56% [7], which is significantly less than the targeted rate of 70% [8]. A high participation rate, however, is necessary to ensure the effectiveness of the MSP, particularly to reduce breast cancer mortality [8].

Because the participation rate in Germany remains too low, despite awareness campaigns and high expenditures used in the invitation process, it is likely that specific reasons exist that lead women to decide against undergoing mammography screenings.

Some predictors of non-participation are already known. Socioeconomic factors, such as education, employment, marital status, and residence (urban/rural), seem to exert a significant impact on MSP participation [9,10,11,12,13,14,15,16]. Women with caregiving responsibilities are equally likely to participate in MSPs as are women without such obligations [17]. Additionally, non-participants exhibited risk factors for other diseases [18]. The ages of the women invited also affect the frequency at which they attend mammography screenings [16,19].

Surveys of participants and non-participants identified important reasons for participation. However, whether participating in other early-detection measures, including mammograms, received outside the screening program, increases or decreases participation rates remain uncertain [19,20,21]. Some studies have shown that the fear of having cancer encourages women to attend mammography screenings [22,23], but other studies have shown the opposite [20,23,24,25]. More reasons associated with non-participation include feelings of indifference [23,26], distrust in preventive measures [25], unavailability of means of transportation [26,27], belief in one’s own health [23], fear of examination results [24] and health restrictions [21,23,25].

Numerous preliminary studies have analyzed measures taken to increase participation rates using complementary educational and motivational interventions [28,29,30,31,32,33,34,35,36,37,38]. Among these measures, telephone consultations are particularly efficient. Additionally, barrier-specific counseling, i.e., information that addresses individual reasons for non-participation, has a higher efficiency than non-specific counseling [39]. Getting counseled could provide a nudge towards screening participation without limiting the individual’s free choice.

We have conducted a randomized controlled trial to assess the effects of barrier-specific telephone counseling and the results have been published [28]. In the course of this study, detailed data on the individuals’ reasons for non-attendance have been obtained.

Little is known about the effectiveness of telephone counseling in relation to individual reasons for non-participation. To the authors’ knowledge, no prospective study has been conducted on this topic. Knowing individual barriers can help make education campaigns more effective and thus improve overall participation rates. Additional studies on the effectiveness of telephone counseling tailored to address individual barriers could help make future interventions more time- and cost-efficient.

This study aimed to determine and quantify the reasons for non-participation in mammography screenings in the Germany-wide MSP and determine whether barrier-specific counseling can influence participation rates.

## 2. Methods

### 2.1. Study Population

The study region comprised one screening center in the state of Mecklenburg-Vorpommern. This rural region has been established as a study region for many years, for example, in the population-based “Study of health in Pommerania (SHIP)” [40]. The population structure is well-defined [41]. The local Ethics Committee of the Ernst-Moritz-Arndt University of Greifswald and the Data Protection Officer of the state of Mecklenburg-Vorpommern, approved the study. Telephone counseling has been shown to significantly increase MSP participation by Hegenscheid et al., 2011 [28].

### 2.2. Inclusion Criteria

Women eligible for the nationwide MSP who failed to have a mammography screening within 6 weeks of a written invitation were included in this study (Figure 1). For the nationwide population-based MSP, all women aged 50–69 years, who have not had a mammogram in the preceding 12 months and have not been diagnosed with breast cancer in the preceding 5 years, are eligible for a biennial mammogram screening. In this study, eligible women were sent a written invitation specifying the location, date, and time for the examination. The examination locations for the eligible women were five radiology institutions and practices that are part of the Greifswald screening unit. All included women had not attended their scheduled screening appointment and had not responded to their reminder letters 6 weeks after delivery. Telephone numbers needed to be available for all included women (Figure 2).

### 2.3. Exclusion Criteria

Women who could not be contacted via telephone because the phone call went unanswered within a 2-week time frame with multiple contact attempts were excluded. Those who did not want to or could not give a reason for their non-participation during the conversation (e.g., because they did not speak German) were also excluded from the intervention group. If the calling consultant found that the client met exclusion criteria for the MSP, that client was also excluded.

The eligible population size was 39,570 women. With a margin of alpha-error of 5% and a power of 80% the sample size was at least 388 for the intervention and 388 for the control group, assuming a 10% higher participation rate in the intervention group. Because the distribution of reasons for non-attendance was not known at the beginning and telephone numbers were thought to be available only for 40% of women, we wanted to include as many women as possible. Due to the efficacy and growing experience of the interviewer as well as the automated handling of the client data, we were able to include 1772 individuals.

### 2.4. Study Procedure

All women who did not respond to the first written invitation received a reminder. This reminder came with a leaflet informing the women that they may be contacted via telephone by a staff member of the University of Greifswald as part of a clinical trial. Randomization to the intervention or control group was achieved using lists provided by the central office for mammography screening. Every second woman on the list was randomized into the intervention group, the other into the control group (1:1). Clients’ phone numbers were retrieved from available data using a computer program (KlickTel 2006, Buhl Data, Burbach, Germany). Women in both groups whose telephone numbers could not be retrieved were excluded. The control group was not contacted via telephone. A total of 1772 women were contacted. Of these, 280 women stated that they had not responded to the invitation because they met exclusion criteria for the MSP. These women were excluded from the study, as were 32 women who did not want to state their reason for non-participation. Overall, 1494 women were interviewed. The interviewer was a female social worker with specialized training in health care counseling, primarily concerning all aspects of the MSP.

All contacted women were asked for consent to telephone counseling at the very beginning of the telephone call. If consent was given, the women received general information about the MSP. After the survey, the clients’ individual reasons for non-participation were determined. The reasons given by the client were categorized by the interviewer using computer software designed for this task (Artemisium GmbH and Co KG). The computer program was designed to record multiple reasons per client. At the beginning of the study, the interview addressed the most common reasons for non-participation, and the software allowed the creation of new categories. If clients’ reasons did not fit any of the existing categories, the interviewer could also type in individual reasons as free text. The free-text reasons were later categorized by two independent examiners. If an individual reason for non-participation did not fit any category it was filed under “other reasons”. If the client stated more than one single reason she was added to multiple subgroups.

#### 2.4.1. Consultation

After determining the reasons for non-attendance, the interviewer provided specific information according to the client’s stated reason. The information was based on official data from the effectiveness and risk/benefit assessment of mammography screening. The trans-theoretical model of Prochaska and DiClemente [42], which was transferred to mammography screening by Rakowski et al. [43,44], was used as a consulting strategy. According to this model, women were informed of the advantages and disadvantages of mammography screenings with the aim of overcoming individual barriers previously preventing them from participating. When the client asked for information on a particular aspect of mammography screening, we used official information issued by the cooperative association for mammography, a non-profit joint venture of the National Association of Statutory Health Insurance Funds and the National Association of Statutory Health Insurance Physicians tasked with certification of all screening units as well as public relations/communication of health information related to the breast cancer screening program [45]. They provided information on the inclusion and exclusion criteria, the conduct of the diagnostic measure, information on pain and discomfort due to mammography, radiation exposure, handling of positive results, consequences of a positive result, and risk assessment. We exclusively used this official information to avoid confusing the client. Telephone calls usually lasted between 15 and 20 min.

#### 2.4.2. Objectives

The analysis was conducted to determine the prevalence of reasons for non-attendance and to investigate which obstacles could be overcome by telephone counseling.

We checked each individual’s attendance 3 months after the telephone consultation using lists from the central screening administration. The evaluation was performed using the database in which individual reasons were recorded.

### 2.5. Statistical Analysis

Statistical analysis was performed in 2022. Table 1 lists the frequencies of patients in the control group and the intervention group, as well as the participation rates. In order to remove inter-categorial influences, we made an adjusted analysis. In an age-adjusted logistic regression, the influence of the variables “Time problems”, “Invitation”, “Date”, “Health-related restrictions”, “Job-related restrictions”, “Unavailability of means of transportation”, “Fear of the examination”, “Fear of the examination result”, “Lack of confidence in the health care system”, “Disbelief in diagnostic validity of mammography”, “Faith in one’s own health”, “Low healthcare utilizer”, “Participates in gray screening” and “Others” were examined in regard to the attendance of mammography-screening.

The reasons for non-participation are given as absolute numbers as well as percentages for participants and non-participants. Categorical data are expressed as the absolute number and percentages. Bivariate associations of potential risk factors with participation were calculated with multinomial logistic regression. Odds ratios (OR) are reported with 95% confidence intervals. A *p*-value of <0.05 was considered to indicate statistical significance.

Statistical analyses were carried out with SAS V9.4 (2002–2012 by SAS Institute Inc., Cary, NC, USA).

## 3. Results

In total, 1494 women were interviewed and counseled via telephone. Allowing for multiple answers, 3281 reasons for non-participation in the MSP were recorded. Whether the client later attended the MSP was determined 3 months post-consultation. The intervention was supported by German cancer aid (Grant no. 107992).

### 3.1. Control Group

The control group consisted of 2486 women. The age structure did not differ from that of the intervention group (mean age control group 59.57 years; mean age intervention Group 59.22 years). All women in the control group were sent written reminders prior to inclusion. Telephone numbers were available for all individuals; however, the controls were not contacted. The participation rate was 28.6% (710 participants; 1776 nonparticipants).

### 3.2. Individual Reasons

We first examined how often specific reasons were stated. We also calculated participation rates for each reason given (Table 1). To determine whether a specific reason for non-attendance had a higher or lower susceptibility to counseling relative to other reasons, we compared participation rates within the intervention group between those who had given that reason and those who had not. Odds ratios (OR) below one suggest that counseling may lead to a higher participation rate, whereas ORs greater than one suggest no effect of counseling. Neither did the number of reasons given by each individual (mean 2.2; median 2.0) correlate to participation, nor did age (data not shown). Odds ratios are shown in Figure 3. For a better overview, individual reasons are shown in four category blocks. These blocks each contain reasons for non-attendance representing a different aspect of mammography screening.

### 3.3. Categories in Detail


*Block 1: Organizational Problems*

*Problems surrounding the organizational aspect of screening*



*Time problems:*


Of the 1494 clients surveyed, 397 (26.57%) stated time problems as a reason why they did not participate in the screening. If the lack of time was due to health-related problems or long working hours, the reason was filed under the respective categories. After consultations, 238 women (59.95%) with time problems attended mammography screening (compared to 40.71% of those who stated other reasons; odds ratio [OR] 0.45; 95% confidence interval [CI] 0.34–0.61; *p* < 0.0001, significant).


*Invitation:*


For 235 women (15.73%), the invitation letter was either not received, ignored, or considered not credible. After telephone contact, 163 of the women (69.36%) appeared for the next scheduled examination (compared to 41.46%; OR 0.32; 95% CI 0.23–0.45; *p* < 0.0001; significant).


*Date:*


The appointment date set in the invitation letter was inconvenient for 215 women (14.39%), and they either did not attempt to change or could not change the appointment date or time. Following a consultation explaining the possibilities for individual scheduling, 69.77% (150 women) participated in the MSP (compared to 41.83%; OR 0.38; 95% CI 0.27–0.52; *p* < 0.0001; significant.)


*Block 2: Limited resources*



*Problems originating in the client’s limited ability to attend*



*Health-related restrictions:*


For 258 women (17.27%), personal health issues or care-taking responsibilities prevented them from attending the screening. One hundred of the advised women decided to participate later (38.76% compared to 47.33%; OR 1.13; 95% CI 0.83–1.54; *p* = 0.4229, not significant).

*Job-related restrictions*:

One hundred eighty-one women (12.12%) stated that they did not attend the screening due to long working hours or difficulty leaving their workplace. Specific counseling enabled 102 of these women (56.35%) to overcome the issues (compared to 44.40%; OR 1.10; 95% CI 0.75–1.61; *p* = 0.6372; not significant).

*Unavailability of means of transportation*:

One hundred eighty-one women (12.12%) stated that they could not arrange transportation to the examination site. After telephone consultation, 66 women (36.46%) attended (compared to 47.14%; OR 1.18; 95% CI 0.83–1.68; *p* = 0.3481; not significant).


*Block 3: Negative perception*



*Barriers resulting from preconceptions*



*Fear of the examination:*


Fear that the examination procedure would be associated with pain or discomfort or that receiving a mammogram could cause cancer or lead to permanent physical damage or fear of radiation exposure prevented 145 women (9.17%) from participating. After consultation, 43 of these (29.66%) received a mammogram (compared to 47.59%; OR 1.34; 95% CI 0.89–2.02; *p* = 0.1628; not significant).


*Fear of the examination result:*


Overall, 122 women (8.17%) said they were afraid of a potentially suspicious finding on their mammogram and therefore did not attend. After a specific consultation, 38 women (31.15%) participated in the screening (compared to 47.16%; OR 1.01; 95% CI 0.69–1.67; *p* = 0.7554; not significant).


*Lack of confidence in the health care system:*


Eighty-four respondents (5.62%) had no confidence in the competence of the doctors or distrusted their motives and believed that mammography screening was dishonest. After consultation, 18 women (21.43%) received the screening (compared to 47.30%; OR 1.72; 95% CI 0.95–3.12; *p* = 0.0730; not significant).


*Disbelief in diagnostic validity of mammography:*


One hundred forty-seven women (9.84%) expressed doubts about the usefulness of mammography screening; they believed that mammographies were either unable to prevent breast cancer or were unnecessary. Forty-four women (29.93%) attended the screening after consultation (compared to 47.59%; OR 1.23; 95% CI 0.74–1.73; *p* = 0.5835; not significant).


*Faith in one’s own health:*


A strong belief in one’s own health prevented 225 women (15.06%) from participating in the screening program; 51 of these women (22.67%) participated in the MSP after telephone counseling (compared to 49.96%; OR 2.01; 95% CI 1.40–2.90; *p* = 0.0002; significant).


*Low healthcare utilizer:*


Seventy women (4.69%) did not want to undergo diagnostic measures in the absence of symptoms. After consultation, nine of these women (12.86%) received mammograms (compared to 47.47%; OR 2.81; 95% CI 1.31–6.01; *p* = 0.0078; significant.).


*Block 4: Other*



*Other reasons*



*Participates in gray screening:*


These are women who had already received diagnostic measures for early detection of breast cancer outside the MSP (e.g., regular mammograms, gynecological examinations, breast ultrasound, self-screening). Over half of all respondents (770 women; 51.54%) indicated this and therefore did not respond to the MSP invitation. After consultation, 354 women (45.91%) participated in the MSP (compared to 45.72%; OR 1.06; 95% CI 0.84–1.33; *p* = 0.5974; not significant).


*Other:*


Two hundred fifty-one women (16.80%) had personal reasons or reasons that did not fit in any other category. Of these, 65 (25.90%) participated in the investigation after consultation (compared to 49.88%; OR 1.70; 95% CI 1.21–2.38; *p* = 0.0020; significant).

## 4. Discussion

### 4.1. Summary

We analyzed reasons for non-participation in the national MSP given in a population-based intervention study and the potential of barrier-specific counseling [46]. Organizational problems have been shown to be most amenable to telephone counseling, whereas the largest group of non-participants, who took gray-screening measures was impervious to counseling.

### 4.2. Context

Numerous studies have shown that, compared to repeated written invitations, telephone consultation can increase participation rates [13,18,19,47]. The current study shows that the effectiveness of this intervention mostly depends on the individuals´ reason for non-attendance. Overall, the participation rate for women who had not complied with the initial written invitation was significantly higher when they were counseled by phone (45.85% vs. 28.56% controls; RR 1.61; 95% CI 1.48–1.74). This rate of increase was consistent with other studies [30,31,33,35,36,39,48] where absolute increases in participation from 10 to 41% were found. Notably, we included only women who were actually advised by phone.

Most of the reasons given for non-participation in MSPs in other studies [19,20,21,22,23,24,25,26,27] were also given by participants in the current study. Most participants (51.54%) indicated that they had taken measures for breast cancer screening outside the national MSP. This is consistent with the findings of Baré in 2003 who found a high proportion of non-participants performing self-scanning and undergoing regular gynecological examinations [19]. In 2014, Moutel et al [49]. showed a connection between low participation rates in organized screening programs and a high number of mammograms outside the MSP. Our findings were consistent with these, likely because of the low-threshold access to mammograms combined with the very high rate of women covered by health insurance in Germany. Because women who had a mammogram in the last year before their scheduled screening were excluded from the study, the actual number of women in the gray-screening category may be higher than the proportion calculated here. This deserves special attention because organized screening programs have higher examination and diagnostic standards than do those performed outside these programs [49].

### 4.3. Key Findings

To our knowledge, this is the first analysis of the influence of telephone counseling according to individual reasons for non-participation (PubMed Search). In this analysis, we compared the ORs of various specific reasons for non-participation in the MSP. We compared the participation rates within a group of counseled women, depending on their stated reasons for non-attendance, as well as with a control group of women who did not receive telephone counseling. The intervention showed particularly good effects among clients who could not attend on the examination date specified in the invitation letter as well as for those who stated that they did not receive the invitation or had perceived it as non-credible. Participation rates also increased in the group that had time problems participating in the screening.

Some groups were not more likely to attend after consultation, including the group that used medical services only in cases of illness (low healthcare utilizer). If the reason for non-attendance was the belief in one’s own health or distrust in the health care system, telephone counseling did rather not alter that individual’s decision. The group that underwent gray screenings also showed no significant increase in participation after counseling.

Our results show that telephone counseling is a successful means of overcoming organizational barriers towards participation in the MSP. In these cases, the opportunity to reschedule their appointment is an effective nudge towards screening participation.

### 4.4. Implications for Future Research

Approaches to address the most common reasons for non-attendance should focus on finding ways to reduce gray screenings and better communicate the option of individual scheduling of MSP appointments. Therefore, technical solutions need to be explored, that address these issues. Moreover, future research should find ways to identify the cause of negative conceptions of mammography screening and ways to make women rethink these.

### 4.5. Strengths and Limitations

The quantification and distribution of reasons for non-attendance had, to the authors′ knowledge, not been studied before. We were able to include a great number of women, who we questioned and counseled in a consistent way.

The current study had some limitations. The reasons for non-participation could be obtained only from women whose telephone numbers were available (35% of all eligible women). Telephone numbers were not part of the data provided in the local registries and had to be retrieved using commercial software. Hence, many women could not be contacted for counseling. The time frame in which clients had to be contacted was limited to 2 weeks, and not all women could be counseled in this time frame. Furthermore, all clients who were asked for their reasons for non-attendance were counseled. Therefore, we could not obtain participation rates for non-counseled women relative to their reasons for non-attendance. The study endpoint was participation within 3 months after the reminders were sent. Hence, we could not show how telephone counseling affects women’s attitudes toward mammography screening over longer terms. The study was conducted in a rural area. Most stated reasons against participation should hold for urban as well as rural areas. The unavailability of means of transportation, however, might be less common in an urban population. All interviews were conducted by a single staff member. Although the interview was scripted, personal biases cannot be ruled out. Lastly, the study mainly provides descriptive data on non-participation while falling short on offering solutions for overcoming most of the barriers that lead to it. We did not compare other methods to increase participation apart from telephone counseling.

## 5. Conclusions

The majority of women who did not attend German MSP stated specific reasons. Depending on their respective reasons for non-attending the susceptibility to counseling varied greatly. Among these reasons, participation in gray screenings was the most common, followed by time problems. Women who reported not having received the invitation or could not make the proposed examination date had the most benefit from telephone consultation. Similarly, issues concerning the proposed examination date could be resolved by counseling. Understanding these reasons may help to address common organizational obstacles and personal reservations.

Moreover, the prevalence of gray screening should be reduced to improve participation in organized mammography screening.

## Figures and Tables

**Figure 1 healthcare-11-00017-f001:**
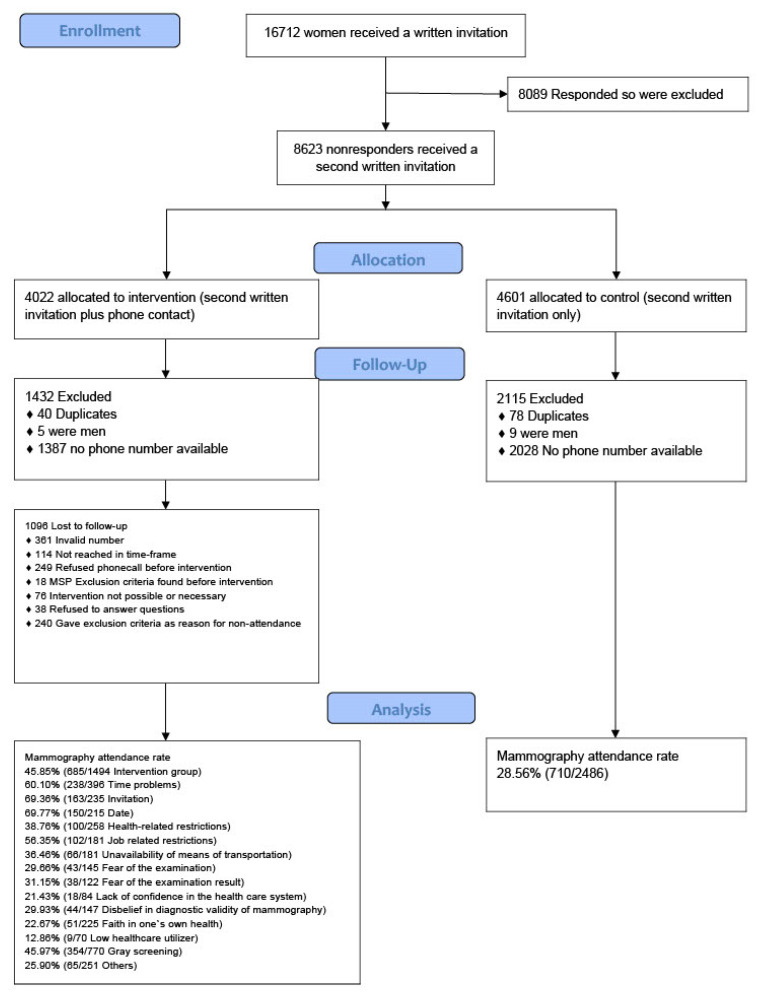
Participant flow chart.

**Figure 2 healthcare-11-00017-f002:**
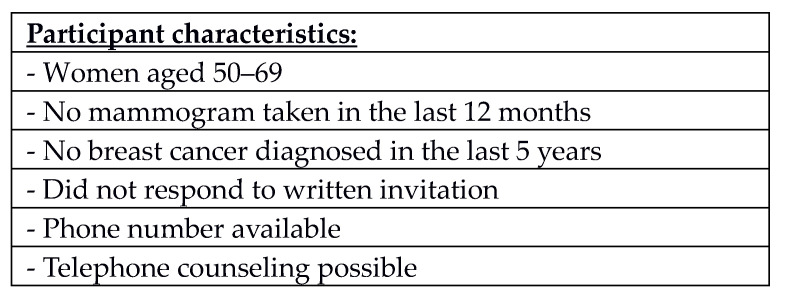
Participant characteristics.

**Figure 3 healthcare-11-00017-f003:**
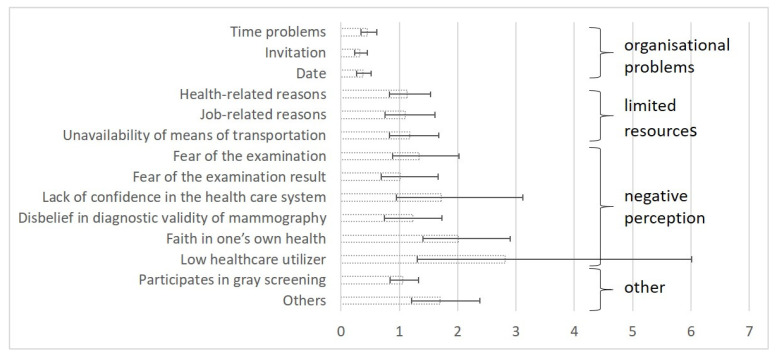
Odds-ratios for participation.

**Table 1 healthcare-11-00017-t001:** Reasons for non-attendance.

	*n*	%	Participants	Non-Participants	Participation Rate	OR	95%CI		*p* Value
Controls	2486		710	1776	28.56%				
Intervention group	1494		685	809	45.85%				
Block 1: Organizational problems									
Time problems	396	26.51%	238	158	60.10%	0.45	0.34	0.61	<0.0001
Invitation	235	15.73%	163	72	69.36%	0.32	0.23	0.45	<0.0001
Date	215	14.39%	150	65	69.77%	0.38	0.27	0.52	<0.0001
Block 2: Limited resources									
Health-related restrictions	258	17.27%	100	158	38.76%	1.13	0.83	1.54	0.4229
Job-related restrictions	181	12.12%	102	79	56.35%	1.10	0.75	1.61	0.6372
Unavailability of means of transportation	181	12.12%	66	115	36.46%	1.18	0.83	1.68	0.3481
Block 3: Negative perception									
Fear of the examination	145	9.71%	43	102	29.66%	1.34	0.89	2.02	0.1628
Fear of the examination result	122	8.17%	38	84	31.15%	1.01	0.69	1.67	0.7554
Lack of confidence in the health care system	84	5.62%	18	66	21.43%	1.72	0.95	3.12	0.0730
Disbelief in diagnostic validity of mammography	147	9.84%	44	103	29.93%	1.23	0.74	1.73	0.5835
Faith in one’s own health	225	15.06%	51	174	22.67%	2.01	1.40	2.90	0.0002
Low healthcare utilizer	70	4.69%	9	61	12.86%	2.81	1.31	6.01	0.0078
Block 4: Others									
Participates in gray screening	770	51.54%	354	416	45.97%	1.06	0.84	1.33	0.5974
Others	251	16.80%	65	186	25.90%	1.70	1.21	2.38	0.0020

## Data Availability

The data presented in this study are available on request from the corresponding author. The data are not publicly available to ensure the protection of personal data.

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
