# Peer review of "Reasons for Non-Attendance in the German National Mammography Screening Program: Which Barriers Can Be Overcome Using Telephone Counseling?—A Randomized Controlled Trial"

_healthcare, 2022, doi:10.3390/healthcare11010017_

Round 1

Reviewer 1 Report

Peer review report

Reasons for non-attendance in a national mammography 1 screening program: Which barriers can be overcome?

This is an interesting read which explores the potential of barrier-specific counselling.  However, the article is poorly structured.  I recommend the inclusion of a study-flow diagram and tabulated outline of participant characteristics to (a) make for a clearer read and (b) free up word count for inclusion of more detail regarding study procedure (participant consent, duration of interviews etc), statistics analysis (currently only significance levels and software used are outlined) and a more comprehensive and streamlined discussion section.  As it stands, the word count is relatively small, so you have plenty of leverage to add more detail and interpretation of results.

Title:  Maybe incorporate ‘which barriers can be overcome’ within first part of the title. It may also be an idea to specify study design (RCT) and target population (non-attenders of German MSP): E.g.  ‘Reasons for non-attendance in Germany’s national Mammography Screening Programme and how barrier-specific counselling can improve attendance:  A randomised controlled trial’.

Introduction

Comprehensive but structure needs improving.

Think about moving lines 56-53 to follow the outline of known reasons for non-attendance.  At present, it sits in the middle!

May be worth incorporating lines 56-53 within paragraph 85-91. 

Methods

Study population: 99-100. Is there any specific reason why Mecklenberg-Vorgpommern is as established ‘study region’ and how is this relevant to mammography screening attendance.  If not, the reader doesn’t need this information.

103: Efficacy of telephone counselling on what outcome?

103-104:  What is unique about Pommerania with regard to the efficacy of telephone counselling?

Inclusion criteria: 117-118:  Remove as this software is referenced under ‘Course of study’ (144:146)

133:  Telephone numbers available for only 40% correctly mentioned as a limitation (370-371) but there you state that only 35% of numbers were available.

134-135:  What do you mean by “Due to great personal resources we were able to include 1772 individuals’?

137: ‘Course of study’ should be changed to Study procedure

It would be useful to include a CONSORT diagram to illustrate participant flow of this RCT (e.g., response rates, exclusions, drop-outs, numbers in each condition and time-frames.

152-153:  Is it correct that only 1 interview interviewed 1494 women?  If so, this should be mentioned as a limitation (e.g., possible source of bias).

156-158:  More information is required to outline categories of non-participated.  What were the pre-determined categories inputted into the software and how were these derived and defined? You mention the software as if it has a specific categorization function.  What is this and what features of this software make it appropriate for categorizing reasons for non-attendance?

Further information required regarding the provision of informed consent, how long the interviews lasted and were participants incentivised in any way?

167–171 :  It might be an idea to include supplementary materials outlining the interview topic guide (ITG).  More information about why the TTM was selected over other health behaviour models (e.g., Theory of Planned Behaviour, Health Belief Model, etc.,).  In other words, what features of the  TTM make it appropriate for your study design and research aims.

176-178:  This sentence doesn’t make sense in this context.  Are you trying to suggest that the consultation procedure was guided by materials provided cooperative association for mammography.  If so, you need to specify why and how?

178:  For international readers like myself, it would be useful to know more about the cooperative association of mammography.  For example, is it specifically patient-centred and a source of information on breast screening accessible to the German public?

Statistical analysis

This section does not adequately describe the methods used to analyse study data.

Results

201-202: The reader doesn’t need to know the cost of the intervention.  Unless an economic evaluation was a sub-aim of the study, the overall cost result per se. 

It might be useful to include a table outlining participant characteristics for both study arms.

230-233:  You need to make it clearer that the OR’s apply to women who cited time-issues as a barrier.

Why are time-issues under ‘Organizational Problems’ not ‘Limited resources’. 

It might be an idea to have a couple of sentences outlining how the 2 blocks and their components are organised and how these related to the literature outlined in the Introduction section.  As it stands, the layout is quite confusing.

A lot of the text included here is repeating what is included in Table 1 and Table 2.

Discussion:

320:  Potential of barrier-specific counselling rather than ‘possibility’

329:  What do you mean by this in relation to other studies?

323-324: Outline the main reasons for non-attendance that were most amenable to barrier-specific counselling within the first paragraph of the discussion sections.  This can be followed up in more detail in subsequent paragraphs.

346-355:  should be incorporated within this first paragraphed or positioned nearer to it.

The discussion should adhere to the following structure:

·         Summarised interpretation of results

·         How the findings relate to previous/ongoing research

·         Possible explanations of results

·         Implications for future research*

·         Strengths and limitations*

* These sections traditionally use these sub-titles. 

Author Response

Peer review report 1

Reasons for non-attendance in a national mammography screening program: Which barriers can be overcome?

Dear Reviewer,

thank you for your review. The authors highly appreciate the work you put into providing feedback to the manuscript.

Your comprehensive and detailed report clearly shows that you are deeply familiar with the topic and the challenges surrounding it.

Therefore the authors found your remarks particularly useful in their effort to improve the article.

We tried to give a point-by-point response to your suggestions. The changes have been incorporated in the revised manuscript.

This is an interesting read which explores the potential of barrier-specific counselling.  However, the article is poorly structured.  I recommend the inclusion of a study-flow diagram

Study flow diagram has been added.

and tabulated outline of participant characteristics

Tabulated outline has been added.

to (a) make for a clearer read and (b) free up word count for inclusion of more detail regarding study procedure (participant consent,

Added line 176-177 “All contacted women were asked for consent to the telephone counselling at the very beginning of the telephone call. If consent was given,”.

duration of interviews etc),

Added line 205 “Telephone calls usually lasted between 15 and 20 minutes.“

statistics analysis (currently only significance levels and software used are outlined) and a more comprehensive and streamlined discussion section.  As it stands, the word count is relatively small, so you have plenty of leverage to add more detail and interpretation of results.

Title:  Maybe incorporate ‘which barriers can be overcome’ within first part of the title. It may also be an idea to specify study design (RCT) and target population (non-attenders of German MSP): E.g.  ‘Reasons for non-attendance in Germany’s national Mammography Screening Programme and how barrier-specific counselling can improve attendance:  A randomised controlled trial’.

Changed Title to “Reasons for non-attendance in the German national mammography screening program: Which barriers can be overcome using telephone counseling? - A randomized controlled trial

Introduction

Comprehensive but structure needs improving.

Think about moving lines 56-53 to follow the outline of known reasons for non-attendance.  At present, it sits in the middle!

May be worth incorporating lines 56-53 within paragraph 85-91. 

We moved the paragraph. Now it reads better. Thank you!

Methods

Study population: 99-100. Is there any specific reason why Mecklenberg-Vorgpommern is as established ‘study region’ and how is this relevant to mammography screening attendance.  If not, the reader doesn’t need this information.

The reason we call Mecklenburg Vorpommern an established study region is because of the large scale comprehensive health study SHIP (Study of health in Pommerania). This study provides extensive data on population-based health in this specific region. We think this could be useful information if you try to compare our results to other regions.

103: Efficacy of telephone counselling on what outcome?

Changed to: “Telephone counseling has been shown to significantly increase MSP participation by Hegenscheid et. al. 2011.”

103-104:  What is unique about Pommerania with regard to the efficacy of telephone counselling?

There is nothing unique about it in this regard. We deleted this part.

Inclusion criteria: 117-118:  Remove as this software is referenced under ‘Course of study’ (144:146)

Removed: “were retrieved using a commercial Software and “

133:  Telephone numbers available for only 40% correctly mentioned as a limitation (370-371) but there you state that only 35% of numbers were available.

Thank you for pointing this out. We made a preliminary assessment on the availability on telephone numbers in order to calculate a sample size. When we made actual phone calls we saw that, due to invalid numbers (mostly outdated telephone numbers), only 35% were available.

We changed the sentence to “ Because the distribution of reasons for non-attendance was not known at the beginning and telephone numbers were thought to be available only for 40% of women...”

134-135:  What do you mean by “Due to great personal resources we were able to include 1772 individuals’?

We changed the sentence to: “The sample size was calculated for 388 individuals in the intervention group. Due to the efficacy and growing experience of the interviewer as well as automated handling of the client-data we were able to include 1772 individuals.”

We hope that clarifies it.

137: ‘Course of study’ should be changed to Study procedure

Changed subtitle to “Study procedure”

It would be useful to include a CONSORT diagram to illustrate participant flow of this RCT (e.g., response rates, exclusions, drop-outs, numbers in each condition and time-frames.

Added consort flow diagram.

152-153:  Is it correct that only 1 interview interviewed 1494 women?  If so, this should be mentioned as a limitation (e.g., possible source of bias).

Yes, that is correct. Added: “All interviews were conducted by a single staff member. Although the interview was scripted, personal biases cannot be ruled out.”

156-158:  More information is required to outline categories of non-participated.  What were the pre-determined categories inputted into the software and how were these derived and defined?

The pre-determined Categories were: gray screening31–33 , fear of the examination result 33,34 Disbelief in diagnostic validity of mammography37,transportation problems38,39 belief in one's own health35 and health related restrictions.33,36,37

The pre-determined categories were derived from the studies mentioned.

The definition for every category is short and can be found under “Categories in detail”.

You mention the software as if it has a specific categorization function.  What is this and what features of this software make it appropriate for categorizing reasons for non-attendance?

It is correct, that the software has a specific categorization function as it was specifically programmed for this study. When the client was asked for her reason for non-attendance, a list of the existing categories was offered to the interviewer. She would then choose one or more categories, depending on the clients individual statement. If the statement did not fit any of the categories, the statement was filed as free text as stated by the individual. If the interviewer found recurring answers from clients, she could then access a function called “add new category”. This category was then shown as a standard option in the next interviews.

Further information required regarding the provision of informed consent, how long the interviews lasted and were participants incentivised in any way?

Added “All contacted women were asked for consent to the telephone counselling at the very beginning of the telephone call. If consent was given,”

Added “Telephone calls usually lasted between 15 and 20 minutes.“

There was no incentive. All clients were informed via letter, that they might get a telephone call by a staff member of the University of Greifswald. They were told that they could decline taking part in the interview, without any repercussions.

167–171 :  It might be an idea to include supplementary materials outlining the interview topic guide (ITG).  More information about why the TTM was selected over other health behaviour models (e.g., Theory of Planned Behaviour, Health Belief Model, etc.,).  In other words, what features of the  TTM make it appropriate for your study design and research aims.

We chose the TTM because it has already been transferred to mammography screening by Rakowski et al.43,44 This fact made it the best applicable model for our study.

176-178:  This sentence doesn’t make sense in this context.  Are you trying to suggest that the consultation procedure was guided by materials provided cooperative association for mammography.  If so, you need to specify why and how?

Yes, that needs clarification. We changed it to:

When the client asked for information on a particular aspect of mammography screening, we used official information issued by the cooperative association for mammography, a non-profit joint venture of the National Association of Statutory Health Insurance Funds and the National Association of Statutory Health Insurance Physicians tasked with certification of all screening units as well as public relations/communication of health information related to the breast cancer screening program.45 They provided information on the inclusion and exclusion criteria, the conduct of the diagnostic measure, information on pain and discomfort due to mammography, radiation exposure, handling of positive results, consequences of a positive result and risk assessment. We exclusively used this official information to avoid confusing the client.”

178:  For international readers like myself, it would be useful to know more about the cooperative association of mammography.  For example, is it specifically patient-centred and a source of information on breast screening accessible to the German public?

Added: “he cooperative association for mammography, a non-profit joint venture of the National Association of Statutory Health Insurance Funds and the National Association of Statutory Health Insurance Physicians tasked with certification of all screening units as well as public relations/communication of health information related to the breast cancer screening program.”

Statistical analysis

This section does not adequately describe the methods used to analyse study data.

Added: In an age-adjusted logistic regression, the influence of the variables “Time problems”, “Invitation”, “Date”, “Health-related restrictions”, “Job-related restrictions”, “Unavailability of means of transportation”, “Fear of the examination”, “Fear of the examination result”, “Lack of confidence in the health care system”, “Disbelief in diagnostic validity of mammography”, “Faith in one’s own health”, “Low healthcare utilizer”, “Participates in gray screening” and “Others” is examined in regard to the attendance of mammography-screening.

Results

201-202: The reader doesn’t need to know the cost of the intervention.  Unless an economic evaluation was a sub-aim of the study, the overall cost result per se. 

We agree. We deleted the information on overall costs.

It might be useful to include a table outlining participant characteristics for both study arms.

Maybe now that the consort flow diagram is in the manuscript, participant characteristics are clearer.

230-233:  You need to make it clearer that the OR’s apply to women who cited time-issues as a barrier.

We added :” After consultations 238 women (59.95%) with time problems attended mammography screening...”

Why are time-issues under ‘Organizational Problems’ not ‘Limited resources’. 

We can see, why that is confusing. The reasoning behind this is, that these time problems could, in most cases, be overcome by offering organizational help, like scheduling a new appointment.

It might be an idea to have a couple of sentences outlining how the 2 blocks and their components are organised and how these related to the literature outlined in the Introduction section.  As it stands, the layout is quite confusing.

We added a sentence explaining each block.

A lot of the text included here is repeating what is included in Table 1 and Table 2.

Yes, it is redundant, but it would seem incomplete to us, if we just had the table.

Discussion:

320:  Potential of barrier-specific counselling rather than ‘possibility’

Yes, that is better. We changed it.

329:  What do you mean by this in relation to other studies?

We changed the sentence to “This rate of increase was consistent with other studies11,12,14,16,17,20,48 where absolute increases in participation of 10%–41% were found.” We hope, that clarifies it.

323-324: Outline the main reasons for non-attendance that were most amenable to barrier-specific counselling within the first paragraph of the discussion sections.  This can be followed up in more detail in subsequent paragraphs.

We added: “Organizational problems have been shown to be most amenable to telephone counseling, whereas the largest group of non-participants, who took gray-screening measures was impervious to counseling. “

346-355:  should be incorporated within this first paragraphed or positioned nearer to it.

The discussion should adhere to the following structure:

·         Summarised interpretation of results

·         How the findings relate to previous/ongoing research

·         Possible explanations of results

·         Implications for future research*

·         Strengths and limitations*

* These sections traditionally use these sub-titles. 

The discussion has been restructured according to your suggestions.

We are grateful for your notes and hope you deem the additions made to the manuscript sufficient.

The revised manuscript is attached to this message.

Sincerely,

Sebastian Fochler

Reviewer 2 Report

The manuscript focuses on identifying main and secondary reasons for non-participation in the national program for screening and mammography. The paper is concentrate to determine what are the barriers that can be overcome in raising participation rates as well as the aspects that help to increase the level of confidence in the target segment to participate in this the program. The paper is in its general form for publication, but it needs a number of necessary amendments before its final acceptance. These observations can be summarized in the following points:

§  The paper is mainly based on the presentation of the descriptive data, as it presents the obstacles that are considered directly affecting the non-participation in the national breast cancer screening program. These data produce many appropriate reasons and require finding direct solutions to solve them, but they do not represent a clear depth in finding solutions to spread the awareness and develop work on it through deep research methodologies in the same field.

§  I think it is important to develop a technical mechanism and methodology to work on revealing the reasons for not participant to deal with awareness programs in the early detection of chronic diseases, and this will help increase the level of acceptance of them, directly or indirectly.

§  Searching for solutions based on the experiences of theoretical frameworks in the field of technology acceptance and adoption, which will help increase the level of theoretical, practical, and developmental knowledge in electronic systems on the one hand, and the knowledge programs directly related to it.

Author Response

Review Report 2

The manuscript focuses on identifying main and secondary reasons for non-participation in the national program for screening and mammography. The paper is concentrate to determine what are the barriers that can be overcome in raising participation rates as well as the aspects that help to increase the level of confidence in the target segment to participate in this the program. The paper is in its general form for publication, but it needs a number of necessary amendments before its final acceptance. These observations can be summarized in the following points:

§  The paper is mainly based on the presentation of the descriptive data, as it presents the obstacles that are considered directly affecting the non-participation in the national breast cancer screening program. These data produce many appropriate reasons and require finding direct solutions to solve them, but they do not represent a clear depth in finding solutions to spread the awareness and develop work on it through deep research methodologies in the same field.

§  I think it is important to develop a technical mechanism and methodology to work on revealing the reasons for not participant to deal with awareness programs in the early detection of chronic diseases, and this will help increase the level of acceptance of them, directly or indirectly.

§  Searching for solutions based on the experiences of theoretical frameworks in the field of technology acceptance and adoption, which will help increase the level of theoretical, practical, and developmental knowledge in electronic systems on the one hand, and the knowledge programs directly related to it.

Dear Reviewer,

thank you for your review an giving us the opportunity to improve the article. The authors highly appreciate the work you put into reviewing the manuscript.

§1 We agree, that the study conducted mainly produces descriptive data while falling short of providing solutions for the problem. Since telephone counseling was the only method, used in this study to increase participation, we can unfortunately only provide data on the effectiveness of this single measure.

We have added a section in „limitations“ (line 433-436) to adress this weakness of the study pointed out by you.

§2 We agree that more research is needed to develop a mechanism to identify barriers keeping patients from taking part in early detection programs. We hope the data provided will help in the development of such measures.

We have added a remark on the need to develop technical solutions in line 415-416

§3 The authors share the optinion of the reviewer, that technology adaption and acceptence would be the most effective means of improving participation and effectiveness in large-scale screening programs. Unfortunately this is not the authors field of expertise. We would be happy to see the data provided be used to develop technology designed to improve screening participation.

We are grateful for your notes and hope you deem the additions made to the manuscript sufficient.

Furthermore the paper has been proof-read and language errors have been corrected. Please consider changing your „English language and style“ rating if you find the changes sufficient.

The revised manuscript is attatched to this message.

Sincerely,

Sebastian Fochler

Reviewer 3 Report

This is a very interesting study, and you have done important work for mammography screening.

I would liked to have seen a discussion both in the introduction and discussion section on the problems on nudging in the work to get people to participate in voluntary screening programs.

Author Response

Peer review 3

Thank you for your review an giving us the opportunity to improve the article. The authors highly appreciate the work you put into reviewing the manuscript.

I would liked to have seen a discussion both in the introduction and discussion section on the problems on nudging in the work to get people to participate in voluntary screening programs.

We agree, that this problem has not been addressed. Therefore we added a section in the introduction, as well as in the discussion section.

We are grateful for your notes and hope you deem the additions made to the manuscript sufficient.

The revised manuscript is attatched to this message.

Sincerely,

Sebastian Fochler